# Development of a Cyclic Voltammetry-Based Method for the Detection of Antigens and Antibodies as a Novel Strategy for Syphilis Diagnosis

**DOI:** 10.3390/ijerph192316206

**Published:** 2022-12-03

**Authors:** Gabriel M. C. Barros, Dionísio D. A. Carvalho, Agnaldo S. Cruz, Ellen K. L. Morais, Ana Isabela L. Sales-Moioli, Talita K. B. Pinto, Melise C. D. Almeida, Ignacio Sanchez-Gendriz, Felipe Fernandes, Ingridy M. P. Barbalho, João P. Q. Santos, Jorge M. O. Henriques, César A. D. Teixeira, Paulo Gil, Lúcio Gama, Angélica E. Miranda, Karilany D. Coutinho, Leonardo J. Galvão-Lima, Ricardo A. M. Valentim

**Affiliations:** 1Laboratory of Technological Innovation in Health (LAIS), Hospital Universitário Onofre Lopes, Federal University of Rio Grande do Norte (UFRN), Natal 59078-970, Brazil; 2Health Sciences Research Unit: Nursing (UICISA:E), The Nursing School of Coimbra (ESEnfC), 3004-011 Coimbra, Portugal; 3Centre for Informatics and Systems of the University of Coimbra (CISUC), Department of Informatics Engineering, University of Coimbra, 3004-531 Coimbra, Portugal; 4Department of Electrical and Computer Engineering, School of Science and Technology, New University of Lisbon, 1099-085 Lisbon, Portugal; 5Department of Molecular and Comparative Biology, School of Medicine, Johns Hopkins University, Baltimore, MD 21218, USA; 6Vaccine Research Center, National Institute of Allergy and Infectious Diseases, National Institutes of Health, Bethesda, MD 20892, USA; 7Postgraduate Program in Infectious Diseases, Federal University of Espírito Santo, Vitória 29075-910, Brazil

**Keywords:** syphilis, *Treponema pallidum*, electrodes, point-of-care method, screening strategy

## Abstract

The improvement of laboratory diagnosis is a critical step for the reduction of syphilis cases around the world. In this paper, we present the development of an impedance-based method for detecting *T. pallidum* antigens and antibodies as an auxiliary tool for syphilis laboratory diagnosis. We evaluate the voltammetric signal obtained after incubation in carbon or gold nanoparticle-modified carbon electrodes in the presence or absence of Poly-L-Lysine. Our results indicate that the signal obtained from the electrodes was sufficient to distinguish between infected and non-infected samples immediately (T0′) or 15 min (T15′) after incubation, indicating its potential use as a point-of-care method as a screening strategy.

## 1. Introduction

Syphilis and other sexually transmitted infections (STIs) are silent epidemics that affect high- and low-income countries worldwide. Although it is a treatable and preventable disease, according to data from the Center for Disease Control and Prevention (CDC), primary and secondary syphilis cases among adults in the USA rose from 7.4/100,000 in 2015 to 14.5/100,000 syphilis cases in 2021, while 149 stillbirths and infant deaths associated to congenital syphilis were reported in 2020. In addition, according to WHO reports, syphilis in pregnancy is the second leading cause of stillbirth worldwide. Furthermore, it may result in prematurity, low birth weight, congenital hearing loss, and bone malformation [1,2,3,4].

The current syphilis screening and laboratory diagnosis strategies include rapid or non-treponemal methods (e.g., VDRL test) that may present false-negative or false-positive results. Additionally, they do not differentiate the exposed child during pregnancy (with detectable maternal antibodies at birth) from those indeed infected by *T. pallidum*. In addition, according to recent recommendations of several national health agencies (i.e., Brazilian Health Ministry, the CDC, and others), following the traditional algorithms for syphilis diagnosis, positive results in screening tests or even negative results with divergent clinical suspicion should be confirmed using additional treponemal methods (such as FTA-ABS, ELISA, or immunoblots) to elucidate the clinical condition of the patient and guide the appropriate therapeutic management [5,6]. Recently, Ceccarelli and colleagues evidenced the challenges of diagnosing asymptomatic cases of neurosyphilis (which can be extrapolated to other items of evidence of tertiary syphilis infection) in co-infected HIV patients due to the serologic failure process [7]. In this sense, developing new strategies for the detection of *T. pallidum* antigens and antibodies simultaneously in the same sample using low-cost platforms with timely result reports may represent a relevant feature to improve the point-of-care laboratory diagnosis of syphilis and other STIs.

Over recent years, screen-printed carbon electrodes have been used as biosensors in a novel tool for quantifying chemokines and antibodies and identifying pathogens using minimally invasive samples in low-cost, rapid testing, and high-specificity methods [8,9,10,11]. In addition, the electrochemical properties of each electrode change the test sensitivity and the immobilization power of the biological baits. Screen-printed carbon (SPC) electrodes are low-cost disposable devices with high surface activity, high electron mobility, and stability that make them useful for antibody immobilization. However, covalent bonding with other biomolecules does not provide stable coatings and does not allow control of antibody orientation. On the other hand, electrodes utilizing screen-printed carbon modified with gold nanoparticles (SPC-GN) allow more potent and more defined binding with biomolecules and improve heterogeneous electron transfer rate, leading to better detection sensitivity [12].

Considering the minimal variations in the detected signal, the computational analysis of reads represents an additional strategy to improve the test sensitivity and avoid imprecise results. The current paper presents the development of an impedance-based method for detecting *T. pallidum* antigens and antibodies as an auxiliary tool for syphilis laboratory diagnosis.

## 2. Materials and Methods

### 2.1. General Properties of the Developed Device

The prototype developed is a low-cost device that constitutes a potentiostat capable of simultaneously controlling electrochemical cells, performing cyclic voltammetry, preprocessing, and sending the results to an external storage and processing device. The device has a touch-sensitive screen, an interface with the user, a rechargeable internal battery, a microcontroller within an integrated system on the chip circuit, and operational amplifiers (in the configuration of subtractor) to regulate the voltages in the electrochemical cells. These features allow the user to control all peripherals and communicate with the external environment via a Wi-Fi network to exchange information with computers that store and post-process the data. In addition, it has an analog-to-digital converter (A/D) responsible for data acquisition, a digital-to-analog converter (D/A), and operational amplifiers to regulate the voltages in the electrochemical cells.

During the voltammetry analysis, the electrodes loaded with biological samples were inserted into independent reading channels. Then, the evaluated parameters (minimum voltage, maximum voltage, time between reading steps, and voltage step size) were normalized across all readings. The resulting voltammetry graph was displayed on the screen, and the data were saved in a CSV file. Figure 1 summarizes the general proprieties of the developed device. The area under the curve (AUC) data are presented in Table 1 (SPC and SPC-GN electrodes without Poly-L-Lysine treatment) and Table 2 (SPC and SPC-GN electrodes with Poly-L-Lysine treatment).

### 2.2. Biological Sample Processing and Quantification of Voltammetric Signal

Infected serum samples were obtained from pregnant women diagnosed with syphilis (presented with a reagent VDRL or positive rapid test) during antenatal care or childbirth and their babies with reagent/positive results (diagnosed with congenital syphilis). Samples obtained from women and their babies that did not present positive/reagent results during the pregnancy or delivery in the screening tests were used as negative controls. All participants signed the individual informed consent terms approved by the Hospital Universitário Onofre Lopes at the Federal University of Rio Grande do Norte (HUOL/UFRN) Ethics Committee (CAAE #10772919.0.0000.5292). All blood samples were collected in a 5 mL dry tube and centrifuged (400× *g*; 10′; room temperature) to obtain serum. An additional non-treponemal test was performed to confirm their serological status. After incubation, the voltammetric signal was quantified immediately (T0′) or for 15 min (T15′).

### 2.3. Electrode Preparation and Immobilization of Biological Baits

Electrodes utilizing screen-printed carbon (SPC; #DRP-11L; Metrohm DropSens, Asturias, Spain) or screen-printed carbon modified with gold nanoparticles (SPC-GN; #DRP-110AUP; Metrohm DropSens) were used as a strategy to detect the voltammetry signal obtained from the biological samples. They were either pre-treated (4 °C, overnight) or not pre-treated with Poly-L-Lysine 0.1% solution (PLL; #P8920; Sigma-Aldrich, St. Louis, MO, USA) to improve the adsorption capacity of biomolecules to the electrode surface. The electrodes were coated using an anti-human serum albumin antibody (#ab10241; Abcam, Cambridge, UK), recombinant *T. pallidum* p47 protein (#ab43055; Abcam), or *T. pallidum* polyclonal antibody (#PA173103; Invitrogen, Waltham, MA, USA) diluted to reach a ratio of 1:100 in PBS 1X (pH 7.4) and incubated at 2–8 °C for 4 h. After this period, all electrodes were washed with PBST solution (PBS 1X pH7.4 plus 0.1% Tween 20) and blocked with 1% BSA (#A7030; Sigma-Aldrich) solution as previously described [13]. All conditions tested resulted in 204 readings—120 using SPC electrodes (60 electrodes previously treated with PLL solution and 60 untreated) and 84 using SPC-GN electrodes (42 previously treated with PLL solution and 42 untreated). All anti-human serum albumin antibody-coated electrodes were read on Channel #1. In contrast, recombinant *T. pallidum* p47 protein-coated electrodes were read on Channel #2, and *T. pallidum* polyclonal antibody-coated electrodes were read on Channel #3 across all samples evaluated.

### 2.4. Data Processing, Batch Normalization, and Interpretation of Results

The last 100 points of each signal were used to test the differences between infected and uninfected samples. We have four batch cases in the analysis: (I) reading T0′; (II) reading T15′; (III) electrodes not treated with PLL solution; and (IV) electrodes treated with PLL solution. The term batch was used to distinguish the respective set of signals to the four cases described above (i.e., I, II, III, and IV). Thus, for each batch, we have 100-point signals, where the number of signals is determined by the number of samples/individuals in each case. For each batch, we determined the global maximum (max(X)) and the global minimum (min(X)). For the selected point signals from each batch, we applied a min–max 0–1 normalization [14], as follows in Equation (1):(1)$$X_{0-1}=\frac{X-\min\left(X\right)}{\max\left(X\right)-\min\left(X\right)}$$

As result of the normalization applied by Equation (1), the global minimum and the global maximum of the normalized data (*X*_0–1_) for each batch are 0 and 1, respectively. After batch normalization, the categories that differentiate channels (1, 2, or 3), the electrode type (SPC or SPC-GN), and the sample type (Infected/Uninfected) were kept.

### 2.5. Statistical Analysis

The statistical analysis was performed using the nonparametric Mann–Whitney U Test. We considered six comparisons (three channels and two different electrodes), so we used a Bonferroni correction at a significance level of α = 0.05 [15].

## 3. Results

### Screen-Printed Carbon Modified with Gold Nanoparticle Electrodes Can Be Used as a Platform to Identify Antigens or Antibodies in Infected Samples

After immobilizing antibodies or *T. pallidum* p47 protein, SPC or SPC-GN electrodes were incubated with human serum samples. After incubation, the cyclic voltammetry analysis was performed immediately (T0′; Figure 2A–H) or for 15 min (T15′; Figure 3A–H). Each sample was submitted for 454 readings for each channel, with an interval of 9.25 ms. However, only the 350-to-454 interval reads presented significant changes in the detected signal obtained from infected and uninfected samples and were considered during all represented analyses.

The data obtained from our initial validation of T0′ readings indicated that SPC electrodes present a similar cyclic voltammetry profile between infected and uninfected samples on anti-human serum albumin, recombinant *T. pallidum* p47 protein, and *T. pallidum* polyclonal antibody-coated electrodes, which was also indicated in the area under the curve (AUC) data (Figure 2A–D). On the other hand, the analysis of results obtained using SPC-GN electrodes revealed an easily distinguishable cyclic voltammetry profile between infected and uninfected samples, which was also represented in the AUC data (Figure 2E–H).

Interestingly, these data were similar to those obtained after 15 min of sample incubation (T15′), indicating that SPC electrodes can only be used to identify infected samples when coated with *T. pallidum* polyclonal antibodies (Figure 3A–D). However, SPC-GN electrodes presented pronounced changes in the cyclic voltammetry profile between infected and uninfected samples in all tested conditions, indicating that these conditions can correctly identify infected samples (Figure 3E–H). In addition, the pre-treatment with PLL did not change the amplitude of the detected signal or the area under the curve obtained from SPC and SPC-GN electrodes incubated with infected or uninfected samples in both reading times (T0′and T15′). Table 1 and Table 2 summarize the serological status of samples used during the initial validation of the device and the AUC data (considering the readings from 350 to 454) obtained using SPC and SPC-GN electrodes, according to the previous incubation period and the biological target tested. Figure 4 summarizes the process from the input of biological samples to the data processing and presentation of results in the device’s screen and its registration in the electronic medical record.

## 4. Discussion

Developing new screening tools to detect syphilis and other STIs is critical to improving primary health care and allowing timely diagnosis and treatment of infected patients. Despite being a curable and preventable condition, over recent decades, syphilis cases have increased worldwide, resulting in a silent epidemic wave of acquired, maternal, and congenital infections [16,17,18].

In this sense, the main reason behind analyzing biological samples from pregnant women and their respective newborns was due to the relevance of the syphilis diagnosis during pregnancy and the appropriate window of opportunity to initiate treatment to prevent congenital transmission. Considering these facts, all positive samples were obtained from pregnant women in different stages of treatment after diagnosis (according to their prenatal care), adding an extra layer of complexity to the performed analysis since we had samples from patients with active syphilis (those newly diagnosed and who would start treatment) and from previously treated patients who still presented a reactivity in the traditional screening methods. Therefore, despite being an initial limitation, the option to carry out this study with only this group of patients was critical in evaluating the accuracy of this methodology as a new tool for screening syphilis using minimally invasive samples.

Currently, the traditional screening methods (such as VDRL or rapid tests) for syphilis diagnosis are based on the detection of antibodies produced during infection. However, these approaches may lead to false-positive or false-negative results, notably during auto-immune diseases or serologic failure processes [7]. Although the testing algorithms recommend carrying out additional tests after a positive result using a screening method, frequently, a positive result in a screening test triggers a chain of multidisciplinary treatments in adults and newborns, resulting in unnecessary hospitalizations and therapies in false-positive cases.

However, as a new potential screening method, it is essential to consider the similarities and differences between the traditional screening methods and the new voltammetry-based tool to detect antigens and antibodies related to syphilis in clinical samples. Among the similarities, all platforms are low-cost and point-of-care-oriented methods. In addition, the time required for the presentation of results after sample incubation (estimated in 15 min) is similar for both the developed tool and the conventional rapid test. Both present the result faster than the VDRL test because they do not require a laboratory infrastructure to be performed and can be applied in an outpatient environment, even by non-specialized professionals.

In parallel, among the differences between these methods, the voltammetry-based approach can present a quantitative result, the result analysis is not subject to human interpretation, and all generated data can be integrated directly with the patient’s electronic medical record. Collectively, these properties indicate that the developed device has potential as a new laboratory screening tool to detect syphilis and other STIs, aiming to improve the currently available methods.

Considering the modern techniques for laboratory diagnosis of complex diseases and the interface between engineering and the development of new tools for more precise diagnosis, the application of cyclic voltammetry-based methods may represent a significant advance in the detection of antigen–antibody interactions using minimally invasive samples. This approach requires the incubation of a minimal volume (e.g., 50 μL) of samples in electrodes coated with antigens or antibodies and detection using potentiostats or similar devices that can detect changes in the electrical conductivity in positive samples. In addition, recent studies evidenced that SPC and SPC-GN electrodes may be helpful as low-cost disposable devices with high surface activity, high electron mobility, and higher binding activity to biomolecules that can be improved by Poly-L-Lysine treatment, reducing the threshold of detection of serological targets [8,9,10,11,12].

## 5. Conclusions

In the present paper, we evaluate the voltammetric signal obtained after detecting antigens and antibodies related to *T. pallidum* using infected and uninfected samples after incubation in SPC and SPC-GN electrodes. Our results indicate that the voltammetric signal obtained using a portable electrode device was sufficient to distinguish between infected and non-infected samples immediately (T0′) or 15 min (T15′) after incubation. Similar to traditional screening methods, these results indicate that this approach can distinguish infected and uninfected samples using two independent biological baits and a constitutive validation. Taken together, this initial validation suggests that this approach may be used as a point-of-care method as a screening strategy for syphilis diagnosis.

## Figures and Tables

**Figure 1 ijerph-19-16206-f001:**
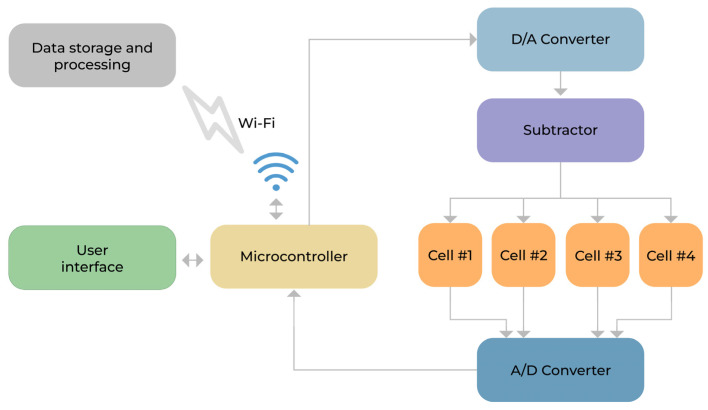
General properties of the voltammetry-based device for detecting antigens and antibodies as a novel tool for syphilis screening.

**Figure 2 ijerph-19-16206-f002:**
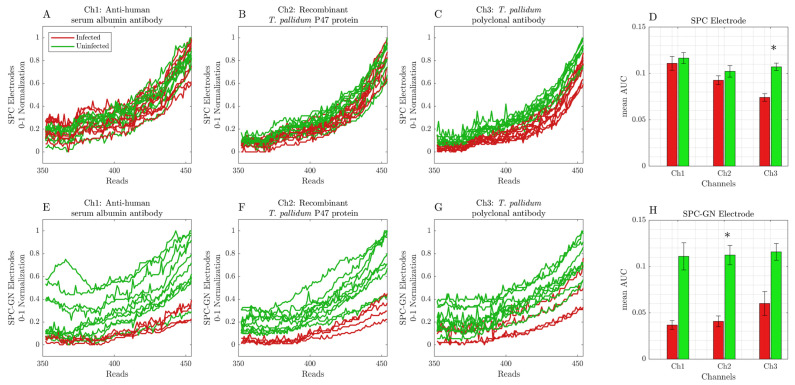
Cyclic voltammetry data were obtained from readings immediately (T0′) after incubating infected and uninfected samples in SPC or SPC-GN electrodes. Each sample was submitted for 454 readings for each channel, with an interval of 9.25 ms. (**A**) anti-human serum albumin (Ch1) in SPC electrodes; (**B**) recombinant *T. pallidum* p47 protein (Ch2) in SPC electrodes; (**C**) *T. pallidum* polyclonal antibody (Ch3) in SPC electrodes; (**D**) AUC data of infected and uninfected samples using SPC electrodes; (**E**) anti-human serum albumin (Ch1) in SPC-GN electrodes; (**F**) recombinant *T. pallidum* p47 protein (Ch2) in SPC-GN electrodes; (**G**) *T. pallidum* polyclonal antibody (Ch3) in SPC-GN electrodes; (**H**) AUC data of infected and uninfected samples using SPC-GN electrodes. The * indicates the differences that resulted statistically significant for the Mann–Whitney U test after Bonferroni correction (at significance level of α = 0.05).

**Figure 3 ijerph-19-16206-f003:**
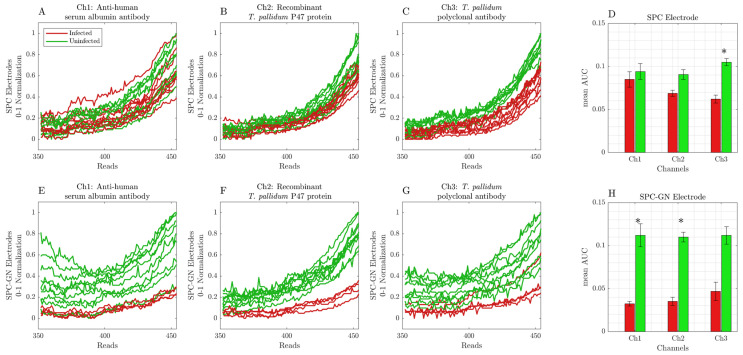
Cyclic voltammetry data were obtained from readings 15 min (T15′) after incubating infected and uninfected samples in SPC or SPC-GN electrodes. Each sample was submitted for 454 readings for each channel, with an interval of 9.25 ms. (**A**) anti-human serum albumin (Ch1) in SPC electrodes; (**B**) recombinant *T. pallidum* p47 protein (Ch2) in SPC electrodes; (**C**) *T. pallidum* polyclonal antibody (Ch3) in SPC electrodes; (**D**) AUC data of infected and uninfected samples using SPC electrodes; (**E**) anti-human serum albumin (Ch1) in SPC-GN electrodes; (**F**) recombinant *T. pallidum* p47 protein (Ch2) in SPC-GN electrodes; (**G**) *T. pallidum* polyclonal antibody (Ch3) in SPC-GN electrodes; (**H**) AUC data of infected and uninfected samples using SPC-GN electrodes. The * indicates the differences that resulted statistically significant for the Mann–Whitney U test after Bonferroni correction (at significance level of α = 0.05).

**Figure 4 ijerph-19-16206-f004:**
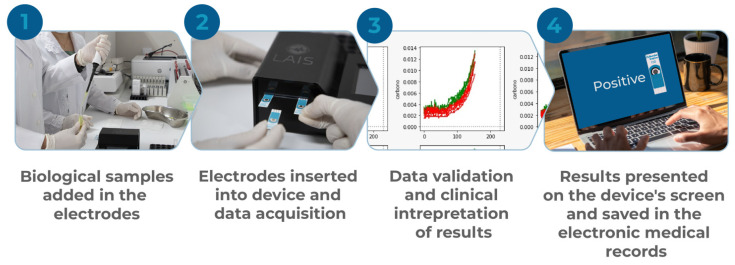
Graphic summary of the procedure from the input of biological samples to the data processing and presentation of results on the device’s screen and its registration in the electronic medical record.

**Table 1 ijerph-19-16206-t001:** Serological status of biological samples and AUC data obtained after incubation with anti-human serum albumin (Ch1), recombinant *T. pallidum* p47 protein (Ch2), and *T. pallidum* polyclonal antibody (Ch3) in SPC and SPC-GN electrodes without Poly-L-Lysine pre-treatment.

Sample ID	Serological Status	Incubation Time	Reading Results					
			SPC Electrodes			SPC-GN Electrodes		
			Ch1	Ch2	Ch3	Ch1	Ch2	Ch3
#1	Reagent (1:16)	T0′	0.0073023	0.0048307	0.0046663	0.0085546	0.0058676	0.0050748
		T15′	0.007248	0.0044141	0.0046389	0.0102268	0.0075225	0.0078614
#2	Reagent (1:8)	T0′	0.0066464	0.0048928	0.0046928	0.0066712	0.0064598	0.0058732
		T15′	0.0070614	0.0055667	0.0049399	0.0082176	0.0067742	0.0072676
#3	Reagent (1:8)	T0′	0.0062958	0.0045392	0.0042912	0.0102056	0.0062748	0.0080925
		T15′	0.0075369	0.0054023	0.0050748	0.0111369	0.0069585	0.009382
#4	Reagent (1:8)	T0′	0.0064392	0.0054418	0.0049408	0.0064092	0.0055343	0.0042261
		T15′	0.0065085	0.0058232	0.0048578	0.0079186	0.0073196	0.0051402
#5	Reagent (1:32)	T0′	0.0066752	0.0050833	0.0050637	0.0054958	0.0049605	0.0063
		T15′	0.0065742	0.0048722	0.0047098	0.0062386	0.0069281	0.0084974
#6	Not reagent	T0′	0.0052951	0.0038428	0.0033974	0.0045085	0.0036376	0.0030425
		T15′	0.0052376	0.0050915	0.0033176	0.0044732	0.0041791	0.0026961
#7	Not reagent	T0′	0.0059196	0.0045167	0.0036095	0.0047245	0.0034605	0.0031814
		T15′	0.0056761	0.0044562	0.0034755	0.0051882	0.0044399	0.0030239
#8	Not reagent	T0′	0.0063853	0.0032065	0.0037765	-	-	-
		T15′	0.0065418	0.0030444	0.0034376	-	-	-
#9	Not reagent	T0′	0.0042196	0.0041471	0.0039121	-	-	-
		T15′	0.0042706	0.0039232	0.0037206	-	-	-
#10	Not reagent	T0′	0.0050824	0.0043886	0.0037699	-	-	-
		T15′	0.0048477	0.0041954	0.0035725	-	-	-

**Table 2 ijerph-19-16206-t002:** Serological status of biological samples and AUC data obtained after incubation with anti-human serum albumin (Ch1), recombinant *T. pallidum* p47 protein (Ch2), and *T. pallidum* polyclonal antibody (Ch3) in Poly-L-Lysine-pre-treated SPC and SPC-GN electrodes.

Sample ID	Serological Status	Incubation Time	Reading Results					
			SPC Electrodes			SPC-GN Electrodes		
			Ch1	Ch2	Ch3	Ch1	Ch2	Ch3
#1	Reagent (1:16)	T0′	0.0061614	0.0050206	0.0054392	0.0053752	0.0115748	0.0060915
		T15′	0.006133	0.0048654	0.0058974	0.0051804	0.0081693	0.0063118
#2	Reagent (1:8)	T0′	0.0071621	0.0045608	0.0055788	0.0179023	0.0073248	0.0043092
		T15′	0.0067964	0.0051062	0.0057716	0.0109739	0.0072356	0.0042765
#3	Reagent (1:8)	T0′	0.0062062	0.0042696	0.0054667	0.0073343	0.0064624	0.0062797
		T15′	0.0058938	0.004919	0.0060235	0.0074324	0.0066493	0.0089889
#4	Reagent (1:8)	T0′	0.0051379	0.0038879	0.0050614	0.012567	0.0063592	0.0050092
		T15′	0.0050918	0.0038203	0.0062732	0.0142846	0.0078542	0.0063647
#5	Reagent (1:32)	T0′	0.0067141	0.0051461	0.0058124	0.0084686	0.0091882	0.0046758
		T15′	0.006716	0.0051173	0.0047245	0.0113376	0.0095065	0.0049608
#6	Not reagent	T0′	0.0076271	0.0055431	0.0038944	0.0054523	0.0037611	0.0037
		T15′	0.0073258	0.0037948	0.0035039	0.0052284	0.0038824	0.0034193
#7	Not reagent	T0′	0.0084239	0.0046605	0.0046275	0.0046131	0.0042807	0.0040147
		T15′	0.0108144	0.0045082	0.0045366	0.0048908	0.0048435	0.0049637
#8	Not reagent	T0′	0.0081389	0.0048268	0.0045644	-	-	-
		T15′	0.0087944	0.0046984	0.0041039	-	-	-
#9	Not reagent	T0′	0.0063222	0.0043167	0.0042317	-	-	-
		T15′	0.0057637	0.0044886	0.0040026	-	-	-
#10	Not reagent	T0′	0.0077984	0.004785	0.0040369	-	-	-
		T15′	0.0083092	0.0047706	0.0041108	-	-	-

## Data Availability

Not applicable.

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
