# Peer review of "Development of a Cyclic Voltammetry-Based Method for the Detection of Antigens and Antibodies as a Novel Strategy for Syphilis Diagnosis"

_ijerph, 2022, doi:10.3390/ijerph192316206_

Round 1

Reviewer 1 Report

Some work has tried to look for diagnostic strategies by algorithms for tertiary forms of syphilis, integrating some work in the 'introduction would make the exposition more complete 

In the materials and metody an iconographic example of the prototype would be helpful to the reader, especially from a clinical background.

You only collected samples from pregnant women.  Could you better motivate why this is so since it might invalidate the methodology of the study. Is it possible to find some clinical information about the patients (perhaps summarized in a table). Also in this regard, the title should be more facetious about the type of patient enrolled.

The discussion should be expanded by citing other work on syphilis with a focus on diagnostic tools or diagnostic strategies (decision trees, algorithms) in other populations as well, such as described in PMID: 31627294

Author Response

  • Comments Reviewer #1:

  1. Some work has tried to look for diagnostic strategies by algorithms for tertiary forms of syphilis, integrating some work in the 'introduction would make the exposition more complete;

Answer: We appreciate this comment. To address this point, we updated the Introduction and Discussion sections presenting data from other authors that explore new diagnostic strategies for syphilis (regardless of progression stage) and other infectious diseases. These alterations are highlighted in the updated manuscript version.

  1. In the materials and metody an iconographic example of the prototype would be helpful to the reader, especially from a clinical background.

Answer: We welcome this suggestion and inform you that we have accepted the recommendation. A graphic summary of the prototype operation was added in the Materials and Methods section, and this alteration is highlighted in the updated manuscript version.

  1. You only collected samples from pregnant women.  Could you better motivate why this is so since it might invalidate the methodology of the study. Is it possible to find some clinical information about the patients (perhaps summarized in a table). Also in this regard, the title should be more facetious about the type of patient enrolled.

Answer: We appreciate this comment. We want to clarify that the current study analyzed biological samples from pregnant women and their respective newborns to evaluate the results obtained using the new voltammetry-based device and compare them with those obtained using traditional laboratory methods (such as VDRL). The main reason to consider this methodology using these samples is related to the importance of diagnosing syphilis during pregnancy. As mentioned in the manuscript, according to the WHO reports, syphilis in pregnancy is the second leading cause of stillbirth worldwide. In addition, it may result in prematurity, low birth weight, congenital hearing loss, and bone malformation [1].

Considering these facts, all the positive samples were obtained from pregnant women in different stages of treatment after diagnosis (according to their prenatal care), adding an extra layer of complexity to the performed analysis since we had samples from patients with active syphilis (those newly diagnosed and who would start treatment) and from previously treated patients but that still presents a reactivity in the traditional screening methods. Finally, all newborn samples were obtained after the beginning of the treatment. Therefore, to preserve the data privacy of enrolled participants, all relevant information for this study was presented in Tables 1 and 2.

In addition, we clarified that this initial study was conducted with a restricted number of participants was critical to evaluate the accuracy of this methodology as a new tool for screening syphilis using minimally invasive samples. Currently, a new testing protocol is running with a higher number of enrolled participants (including male adults and non-pregnant women) to explore this methodology in a diverse population.

  1. The discussion should be expanded by citing other work on syphilis with a focus on diagnostic tools or diagnostic strategies (decision trees, algorithms) in other populations as well, such as described in PMID: 31627294

Answer: We welcome this suggestion. As previously mentioned, we updated the Introduction and Discussion sections with new data from other authors exploring new diagnostic strategies for syphilis (regardless of progression stage) and other infectious diseases. All changes are highlighted in the updated manuscript version.

Reviewer 2 Report

Development of a cyclic voltammetry-based method for the detection of antigens and antibodies as a novel strategy for syphilis diagnosis

The authors presented a novel approach for the diagnosis of sexually transmitted infectious diseases. Overall, the article is well defined and structured. However, some revisions are required.

1)      Lines 37–72; Please mention the techniques previously used for syphilis diagnosis.

2)      Lines 77–83; The author need to provide graphical design of the device.

3)      Lines 91–96; Please mention total sample size.

4)      Lines 104–120; Please provide graphical abstract of the whole procedure.

5)      Lines 122–128; This reviewer didn’t understand the equation, please describe the equation clearly.

6)      Lines 136–161; The author incubated the electrodes for 4 hours, if possible please mention the results at different incubation intervals such as after 30 minutes of incubation until 4 hours.

7)      The reviewer is greatly impressed from the authors approach, but this data is too short. The study can become more impressive, if author use samples from patients of different STIs.

8)      Lines 181–201; Author need to discuss the previously described methods or techniques for the identification of syphilis. Also, mention time difference that how your approach rapidly identified the syphilis compared to traditional approaches?

Author Response

  • Comments Reviewer #2:

Development of a cyclic voltammetry-based method for the detection of antigens and antibodies as a novel strategy for syphilis diagnosis

The authors presented a novel approach for the diagnosis of sexually transmitted infectious diseases. Overall, the article is well defined and structured. However, some revisions are required.

  1. Lines 37–72; Please mention the techniques previously used for syphilis diagnosis.

Answer: We appreciate this suggestion, and we mentioned in lines 48 – 49 the traditional methods used for syphilis screening (VDRL and rapid tests). However, we took the opportunity and included additional information from the Brazilian Health Ministry and the CDC about the confirmatory testing protocol using other treponemal methods (such as FTA-ABS, ELISA, or immunoblots) to elucidate the clinical condition of the patients [2,3]. All changes are highlighted in the updated manuscript version.

  1. Lines 77–83; The author need to provide graphical design of the device.

Answer: We welcome this suggestion. As mentioned to Reviewer #1, to address this point, we add a new figure evidencing the general features of the developed device. This alteration is highlighted in the updated manuscript version.

  1. Lines 91–96; Please mention total sample size.

Answer: We appreciate this question, and, as evidenced in Table 1 and Table 2, 10 biological samples (serum) were used to analyze SPC electrodes without previous treatment with Poly-L-Lysine. 7 biological samples (serum) were used to analyze SPC-GN electrodes without previous treatment with Poly-L-Lysine. In parallel, 10 biological samples (serum) were used to analyze SPC electrodes previously coated with Poly-L-Lysine. In comparison, 7 biological samples (serum) were used to analyze SPC-GN electrodes previously coated with Poly-L-Lysine. As mentioned in section 2.3 (Electrodes preparation and immobilization of biological baits), in all experimental conditions tested, the electrodes were coated using an anti-human serum albumin antibody (#ab10241; Abcam), recombinant T. pallidum p47 protein (#ab43055; Abcam), or T. pallidum polyclonal antibody (#PA173103; Invitrogen) diluted 1:100 in PBS 1X (pH 7.4) and incubated at 2o – 8o C for 4 hours. In summary, we analyzed 204 experimental conditions - 120 using SPC electrodes (60 electrodes previously treated with PLL solution and 60 untreated) and 84 using SPC-GN electrodes (42 previously treated with PLL solution and 42 untreated). We add an extra comment evidencing this data in the Materials and Methods section (lines 100 – 103). This change is highlighted in the updated manuscript version.

  1. Lines 104–120; Please provide graphical abstract of the whole procedure.

Answer: We appreciate the suggestion. To address this point, we add a graphic summary of the procedure since the input of biological samples to the analysis of results. This alteration is highlighted in the updated manuscript version.

  1. Lines 122–128; This reviewer didn't understand the equation, please describe the equation clearly.

Answer: We appreciate the comment. To improve the understanding of equation 1, we insert more details about the data normalization process and how this information is correlated to data analysis and interpretation of results. This information was inserted in section 2.4 (Data processing, batch normalization, and interpretation of results), and all inserts are highlighted in the updated manuscript version.

  1. Lines 136–161; The author incubated the electrodes for 4 hours, if possible please mention the results at different incubation intervals such as after 30 minutes of incubation until 4 hours. 

Answer: We appreciate the comment, but we need to clarify some aspects of the methodology of the antibody or antigen adsorption protocol. The period of incubation with antigens or antibodies in the SPC or SPC-GN electrodes was adjusted considering a similar protocol previously described [4]. This period is required for antigen or antibody immobilization on the surface of electrodes and is performed before sample incubation and data acquisition. All electrode readings were performed immediately (T0’) or 15 minutes (T15’) after incubation with the biological samples (Figures 1 and 2, respectively). Considering these elements, we believe this request did not apply to the current paper.

  1. The reviewer is greatly impressed from the authors approach, but this data is too short. The study can become more impressive, if author use samples from patients of different STIs.

Answer: We would like to thank the comment, and in fact, our data are quite encouraging, although they were obtained using an initially limited number of clinical samples. As mentioned to Reviewer 1, this initial limitation was due to the evaluation of this methodology's accuracy as a new tool for screening syphilis using minimally invasive samples. We also agree that this approach has the potential as a new laboratory screening tool for syphilis and other STIs to improve the currently available methods. Currently, we have a new testing protocol running with a higher number of enrolled participants (including male adults and non-pregnant women) to explore this methodology in a diverse population for HIV, syphilis, and other STIs screening. There are preliminary data that should be presented in future papers while we carry out the entire validation process of this methodology.

  1. Lines 181–201; Author need to discuss the previously described methods or techniques for the identification of syphilis. Also, mention time difference that how your approach rapidly identified the syphilis compared to traditional approaches?

Answer: We appreciate the suggestion and have included additional elements in the discussion of the updated manuscript version exploring the similarities and differences between the current traditional screening methods and the new voltammetry-based tool to detect antigens and antibodies related to syphilis in clinical samples. Among the main similarities we can highlight, the detection time between this new tool and the conventional rapid test is similar (15 minutes between incubating the samples and obtaining the results). In addition, both present the result faster than the VDRL test because they do not need a laboratory infrastructure to be performed and can be applied in an outpatient environment, even by non-specialized professionals. Among the main differences between these methods, the voltammetry-based approach can present a quantitative result, the result analysis is not subject to human interpretation, and all generated data can be integrated directly with the patient's electronic medical record. All changes are highlighted in the updated manuscript version.
